# Learning Time-Invariant Representations for Individual Neurons from Population Dynamics

**Lu Mi**[1,2,*]**, Trung Le**[2,*]**, Tianxing He**[2]**, Eli Shlizerman**[2]**, Uygar Sümbül**[1]
[1] Allen Institute for Brain Science
[2] University of Washington
{lu.mi,uygars}@alleninstitute.org
{tle45, shlizee}@uw.edu
goosehe@cs.washington.edu

## Abstract

Neurons can display highly variable dynamics. While such variability presumably supports the wide range of behaviors generated by the organism, their gene expressions are relatively stable in the adult brain. This suggests that neuronal activity is a combination of its time-invariant identity and the inputs the neuron receives from the rest of the circuit. Here, we propose a self-supervised learning based method to assign time-invariant representations to individual neurons based on permutation-, and population size-invariant summary of population recordings. We fit dynamical models to neuronal activity to learn a representation by considering the activity of both the individual and the neighboring population. Our self-supervised approach and use of implicit representations enable robust inference against imperfections such as partial overlap of neurons across sessions, trial-to-trial variability, and limited availability of molecular (transcriptomic) labels for downstream supervised tasks. We demonstrate our method on a public multimodal dataset of mouse cortical neuronal activity and transcriptomic labels. We report $> 35\%$ improvement in predicting the transcriptomic subclass identity and $> 20\%$ improvement in predicting class identity with respect to the state-of-the-art.

## 1  Introduction

Population recordings of neuronal activity enable relating behaviorally-relevant dynamics to the summary activity of the recorded population. While this has produced numerous insights into how the brain works [1], the activity and identity of individual neurons should be analyzed to achieve a mechanistic understanding at the implementation level [2], which may hold the key to new biologically-inspired algorithms [3, 4]. Moreover, emerging experimental evidence suggests that neurons have diverse yet stable molecular identities, which can dictate their computational roles [5, 6].

Joint (i.e., multimodal) profiling and alignment of electrophysiological features and gene expression of individual neurons suggest a good correspondence between these two modalities in slice experiments [7–11]. Recently, population recordings of calcium activity followed by spatially registered single-cell transcriptomic recordings enabled similar joint profiling of *in-vivo* activity and molecular identity [6]. Importantly, such activity depends on both the intrinsic physiological properties of neurons and the exogenous inputs (synaptic and modulatory) to those neurons, which are themselves a product of both sensory inputs to the organism and the recurrent activity in the brain.

While recording from molecularly defined neuron populations has been a popular method, these experiments do not allow for studying the concurrent responses of different neuron types to stimuli.

---

*Equal contribution

37th Conference on Neural Information Processing Systems (NeurIPS 2023).

On the flip side, joint profiling of panneuronal population activity and transcriptomics is slow, expensive, and not available to many research labs. Learning the association between these two observation modalities can minimize the need for joint profiling and provide neurobiological insights.

The constancy of neuronal identity in adults in the face of a potentially rapidly changing environment represents a key challenge: the inferred identity should be invariant to time and the task that the organism engages with, suggesting that the inference method should ideally be invariant to those variables. In the absence of *a priori* information on identity, it is also desirable that the invariance extends to the number and the (arbitrary) ordering of experimental population. Moreover, a technical challenge common to many multimodal datasets is that only a relatively small fraction of the observations tend to be jointly characterized (or otherwise labeled), limiting the applicability of supervised approaches.

To address these problems, here, we develop a self-supervised approach – Neuronal Time-Invariant Representations (NeuPRINT), to infer neuronal identity from population recordings by forming a model of activity dynamics that depends only on past activity of the neuron itself and statistics of past population activity that are invariant to the ordering of the individuals and asymptotically invariant to the size of the population. We demonstrate the utility of the inferred identities by reporting the performance of a simple classifier of transcriptomic identity on those representations and other baselines. We also study the impact of providing similarly invariant yet more detailed information on the population by partitioning it into center vs surround subsets, reflecting a well-known yet simple connectional and functional property of neuronal circuits [12–14].

## 2 Related work

**Latent dynamical models:** Latent modeling of time series of population activity aims to gain insight on the dynamics of the totality of the activity of a single experimental trial [1, 15–20]. When interpretability of the latent dynamics is not a priority, powerful recurrent neural networks uncover high-fidelity representations [21, 22]. Recently, low-dimensional time series representations were obtained with high reproducibility, albeit without a model of the dynamics, using a contrastive loss objective [23]. Another recent study obtained parametric representations of individual neurons in population recordings within a linear dynamics framework, where the activity of the neuron is informed by a latent representation of the population so that the results are not transferable across studies that do not have the same population with the same ordering of the neurons [24]. The idea of informing neuronal activity prediction of neighboring activity and/or extrinsic covariates such as stimuli or behavior was also explored in previous classical studies [25, 26].

Liu *et al.* [27] proposed the use of the transformer architecture [28] to generate representations of individual neurons in a supervised paradigm, which do not interact with the rest of the population. Instead, those representations are combined to accomplish the supervised task.

**Multi-modal neural data:** Characterization of intrinsic electrophysiology of neurons typically uses a set of features [7, 9, 29, 10, 11] or mechanistic model parameters [30], instead of time series models, to align gene expression and physiology in multimodal slice recordings. Such experiments study only the neuron in isolation and the recordings may not reflect in-vivo activity in the context of a behavior.

A large-scale study that focuses on the *in-vivo* activity of the individual neurons only, and not on their interaction, uses a neural network with attention layers in a supervised setting to predict the neuronal subclass, based on transgenic mouse lines [31].

**Other predictive models:** Implementing nonlinear autoregressive models with exogenous inputs (NARX models) using recurrently connected neural networks has a long history [32]. Modern implementations typically use gated units to address the vanishing gradients problem [33]. Recently, the transformer architecture [28], has produced impressive results in sequence-to-sequence tasks. Despite interpretability issues, such models can be significantly more powerful than classical dynamical systems models (e.g., linear dynamical system) [34, 35]. In this work, we use transformer as a dynamical model to integrate with both individual and population dynamics, and the representation of individual neuron is assigned with a time-invariant embedding.

**Phenomenological Hodgkin-Huxley equation:** For point neuron models, which ignore the spatial distribution of the different ion channels, the celebrated Hodgkin-Huxley (HH) equations provide a phenomenological description of the neuronal membrane potential via coupled differential equations [36, 37]. The membrane potential controls the generation of spiking outputs and calcium

imaging provides an indirect measurement of this voltage over time [38]. For *in vivo* measurements, the synaptic input current injected by other neurons (pre-synaptic partners of the neuron of interest) is represented as exogenous input, which can be written as a sum over the synapses. The HH equations also include multiple neuron-specific, time-invariant parameters. Thus, the HH equations consider neuronal activity as a function of both the activities of other neurons and the intrinsic parameters of its electrophysiology.

# 3 Methods

**Implicit dynamics models for neuronal activity:** Neuronal dynamics are significantly more complex than the HH equations because the physical distribution of the various channels on the neuronal arbor, nonlinear computing abilities of the dendrites, modulatory communication between neurons,etc. can (i) modulate the neuron-specific parameter set, (ii) add more parameters to that set, and (iii) change the functional form of HH equations. Moreover, the relationship between the membrane voltage and the observable of most neuronal population activity experiments, the calcium dynamics, is itself complex. Finally, it appears impossible to perform the detailed measurements needed to fit the parameters of the HH equations based on *in vivo* experiments with current technology. Thus, we pursue an implicit modeling approach while trying to capture the fundamental dependencies with the help of flexible deep neural network parametrization. Let $X_t^{(i)}$ denote the calcium activity of neuron $i$ at time $t$ and consider the following equation for dynamics:

$$\frac{dX_t^{(i)}}{dt} = f(X_t^{(i)}, \bar{P}_t^{(-i)}, \Phi^{(i)}), \tag{1}$$

where $\bar{P}_t^{(-i)}$ denotes the activity of all the neurons that provide (synaptic or extra-synaptic) input to neuron $i$ at time $t$. $\Phi^{(i)}$ denotes a time-invariant representation for neuron $i$, which implicitly captures the intrinsic parameters of HH equations and other such time-invariant aspects of neuronal identity.

**Permutation-invariant summary of population activity:** Even if neurons were identifiable in population imaging experiments, the connectivity of neurons and our observations of it can be considered as stochastic events [39]. Therefore, to enable the transfer of knowledge across sessions, experiments, and individuals, it is highly desirable to approximate the dependency of neuronal dynamics on $\bar{P}_t$ (Eq. 1) in a way that is invariant to permutations, number, and detailed identity of the neurons contributing to it. To achieve this, we propose to replace $\bar{P}_t$ with multiple (asymptotically) invariant statistics of the activity of a neighboring population of neurons, such as average activity [40].

Behavioral observations, such as pupil diameter, can serve as indirect readouts on the activity of unobserved neurons. Concatenating these observations with the aforementioned statistics will enrich the exogenous input observed by the dynamical model. We call this new concatenated variable $P_t$.

**Center-surround partition:** Synaptic connection probability between neurons depends on distance [12, 13]. Similarly, neuronal co-variability correlates with spatial distance [14]. To include this neurobiological insight while maintaining permutation-invariance of our model, we propose a simple extension: we partition the population activity into two groups, center and surround, and compute the relevant invariant statistics for each group separately. Such partitioning is easy to obtain in calcium imaging experiments since the distances between the somata of neurons are readily available. Beyond its simplicity, this choice is also motivated by surround suppression being a connectivity motif in the brain [41, 42].

**Discrete-time dynamical model of neuronal activity:** We re-write neuronal dynamics in discrete time as

$$X_{t+1}^{(i)} = f(X_{t-W+1:t}^{(i)}, (C_{t-W+1:t}^{(-i)}, S_{t-W+1:t}^{(-i)}, B_{t-W+1:t}), \Phi^{(i)}), \tag{2}$$

where $X \in \mathbb{R}^{N \times T}$ represents the activity data for the whole recorded population with $N$ neurons and $T$ time steps. $C$, $S$, and $B$ represent $D$, $D$, and $D$'-dimensional permutation- and size-invariant (i.e., $N$) surrogates for brain activity at each time point. $C$ and $S$ compute identical statistics of population activity (here, mean and standard deviation) except that the statistics in $C$ are calculated over the center partition (neurons whose distance to neuron $i$ is at most $\Delta$) and the statistics in $S$ are calculated over the surround partition (neurons whose distance to neuron $i$ is larger than $\Delta$), see Appendix E.1. Here, $\Delta$ is a hyperparameter. $B$ denotes the contribution of time-resolved behavioral observations. Hence, the triplet $(C_t, S_t, B_t)$ corresponds to $\bar{P}_t$. $\Phi^{(i)} \in \mathbb{R}^K$ denotes a $K$-dimensional

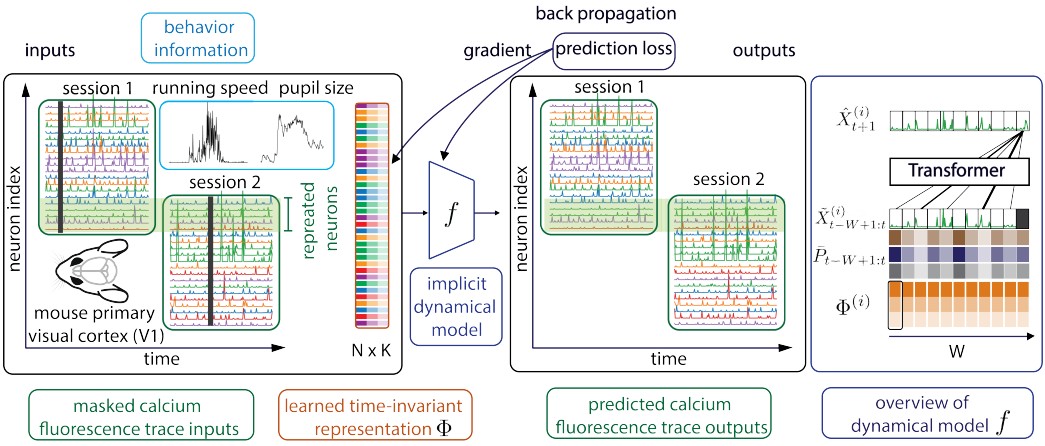

Figure 1: Overview of self-supervised representation learning framework NeuPRINT. Activities of $N$ neurons (recorded by 2-photon calcium imaging of the mouse primary visual cortex) and behavior information (pupil size, running speed, etc.) across multiple sessions are used as inputs to fit an implicit dynamical model $f$ and learn time-invariant $N \times K$ representation $\Phi$. The learned representations are later evaluated on supervised downstream tasks to predict transcriptomic class and subclass identities. In the optimization framework, neuron-specific representation $\Phi_i$ is repeated at every time step, then concatenated with masked past neuronal activity $\tilde{X}^{(i)}_{t-W+1:t}$ and permutation-invariant population inputs $\bar{P}_{t-W+1:t}$ to form the input. The transformer model is trained to predict neural activity $\hat{X}^{(i)}_{t+1}$ at the masked step with a causal attention mask over the $W$-step context window.

time-invariant representation for neuron $i$, and $W$ denotes the width of the available temporal context. The subscripts denote the limits of the time interval within which the corresponding variable is available to $f$.

**Self-supervised Representation Learning Framework with Transformer:** We thus propose to solve the following self-supervised optimization problem to infer both the function $f$ and $\Phi^{(i)}$:

$$\arg \min_{f, \{\Phi^{(i)}\}_i} \sum_{i,t} \mathbb{E}_{X^{(i)}_{t+1}} ||X^{(i)}_{t+1} - f(X^{(i)}_{t-W+1:t}, (C^{(-i)}_{t-W+1:t}, S^{(-i)}_{t-W+1:t}, B_{t-W+1:t}), \Phi^{(i)})||, \quad (3)$$

where $|| \cdot ||$ denotes a norm. It is worth pointing out that $f$ depends on the neuron of interest or other neurons in the population only through its explicit parameters. The reason for this choice is to maintain the transferability and ubiquity of the learned model $f$, the first argument of the optimization, while summarizing neuronal variability with $\Phi^{(i)}$, the second argument.

The idea of inferring invariant representations for neurons directly from *in vivo* recordings by fitting a dynamical model (i.e., predicting activities in the next time step) is reminiscent of, and motivated by, the recent spectacular successes of "foundation models" in natural language modeling [43], where capturing the dynamics of a complicated system produces an implicit understanding of the dynamics and identity of its components. This procedure has been shown useful for multiple downstream tasks [43]. Therefore, we use a transformer model [28] to parametrize the function $f$. Unlike previous uses of the transformers for neural data [34, 35], the input tokens in our model do not come from a countable set because raw calcium recordings are best represented by real valued signals.

To train the transformer and the time-invariant representation, we first generate masked activity inputs $\tilde{X}^{(i)}_{t+1}$, where neuronal activity at time $t + 1$ is zero-out. This masked activity and the past activities $\tilde{X}^{(i)}_{t-W+1:t}$ are concatenated with the time-invariant representations $\Phi^{(i)}$ and permutation-invariant summary of population dynamics $\bar{P}_{t-W+1:t}$ to form the inputs to the transformer. We ask the transformer to predict the activity at the masked step, and compute the loss from the predicted activity $\hat{X}^{(i)}_{t+1}$ and ground truth activity $X^{(i)}_{t+1}$. The dynamical model $f$ and time-invariant representation $\Phi^{(i)}$ are jointly learned during the optimization. To perform this task, the transformer has to learn the temporal progression of neuronal activity conditioned on the neuronal identity and the causal temporal

context of individual and population statistics. An overview of NeuPRINT learning framework is depicted in Figure 1.

## 4 Experiments

### 4.1 A Multimodal Dataset

We use a recent, public multimodal dataset to train and demonstrate our model: Bugeon *et al.* [6] obtained population activity recordings from the mouse primary visual cortex (V1) via calcium imaging, followed by single-cell spatial transcriptomics of the tissue and registration of the two image sets to each other to identify the cells across the two experiments. 2-photon calcium imaging recordings were obtained with a temporal sampling frequency of 4.3Hz. And the spatial coordinates of recorded neurons are also provided. We first evaluate our approach on one animal (SB025) across 6 sessions. The recordings from this animal include 2481 neurons in total. We then extend our analysis on functional recordings from 4 mice (SB025, SB026, SB028, SB030) across 17 sessions. They contain 9728 neurons in total. Each session lasts about 20 minutes and records about 500 neurons. A small subset of neurons overlap across sessions. The subsequent transcriptomic experiment profiles mRNA expression for 72 selected genes in *ex vivo* tissue. These genes were used to identify the excitatory vs inhibitory class labels of neurons. In addition, $51\%$ of the neurons in the inhibitory class of SB025 also have identified subclass labels (Lamp5, Pvalb, Vip, Sncg, Sst). Finally, the dataset includes behavioral information for the mice (running speed and pupil size) during the *in-vivo* recording as well as an assignment for each image frame (i.e., time point) that we call frame state from the set {running, stationary desynchronized, stationary synchronized}.

### 4.2 Benchmark Evaluation for Transcriptomic Identity Prediction

We use the aforementioned public dataset to introduce a new two-step benchmark: (i) self-supervised learning of time- and permutation-invariant representations for individual neurons from population activity, (ii) prediction of labels for each individual neuron based on those representations.

We introduce a downstream classification task to predict the subclass label with supervised learning, where the neurons with subclass labels from all sessions are randomly split into train, validation and test neurons with a proportion of $80\% : 10\% : 10\%$. We further introduce another supervised downstream classification task to predict the class identity only (i.e., excitatory vs inhibitory). In this task, the validation and test neurons in the subclass prediction task are used as validation and test neurons for inhibitory neurons, and the same fraction of excitatory neurons are randomly selected as validation and test neurons. The rest of the recorded population from all sessions is used for training.

We first optimize the dynamical model ($f$ in Eq. 2) and the time-invariant representation on the training set. We use past activities of the training neurons and permutation-invariant summary of population dynamics including pupil size, running speed, frame state, the mean and standard deviation of population activity, and the mean and standard deviation of center-surround activity to predict the individual neurons' activity in the next time step. After training, we fix the dynamical model $f$ and only optimize the time-invariant representations $\Phi$ for training, validation, and test neurons under the same self-supervised learning framework.

Following self-supervised optimization, in the second step, we evaluate the learned representation with two supervised downstream tasks (class and subclass prediction) and three simple classifiers including k-nearest neighbor (KNN), linear model, multi-layer perceptron (MLP) with one hidden layer. The training neurons' representations $\Phi$ are used to train the classifier, and validation neurons are used to tune the hyperparameters (learning rate, hidden dimensionality, number of epochs, etc.), and the top-1 accuracies of all models are reported on the test neurons (Table 1). See Appendix for an analysis of sensitivity.

### 4.3 Implementation of a spectrum of implicit dynamical models and downstream classifiers

We explore four different implicit (not mechanistic) dynamical models: linear, nonlinear, gated-recurrent network (GRU), and transformer with self-attention. We optimize the parameters of the dynamics $f$ and the neuronal representation $\Phi$ using gradient descent for all models.

**Linear model:** We use a linear dynamical system where the activity at the last step is predicted from a linear combination of the activities and statistics of the population activities, behavioral information from previous steps inside a temporal window and the time-invariant representation (which is repeated

at each step). This corresponds to an autoregressive model with exogenous inputs, where statistics of the activities of other neurons and behavioral information constitute the exogenous input to the dynamics of the neuron of interest.

**Nonlinear model:** In addition to the linear model, a nonlinear activation was applied to the weighted activity, behavioral information, repeated neuronal representation at each step before the linear combination (i.e., nonlinear autoregressive model with exogenous inputs).

**Recurrent network with gated units:** The activity at the next step is predicted from the hidden state in addition to the activity at the current step and the repeated neuronal representation as the inputs.

**Transformer:** We implement a $W$-step causal attention mask such that the transformer predicts the activity of the neuron at the current time step based on the hidden states in the $W$ previous time steps. The hidden state tasks the neuron activities, statistics of the population activities, behavioral information at that time step, and time-invariant representation repeated at each time step as inputs. We use the transformer encoder-only implementation from PyTorch [44] with 2 attention heads.

**Training details:** For the objective function to predict the activity, we explore both mean squared error (MSE) and negative log likelihood (NLL) with a Gaussian distribution. To train the dynamical model and representation of neurons, we use a 64-dimensional embedding for the time-invariant representation. The temporal trial window size is 200 steps for the linear, nonlinear models, recurrent network and transformer. The batch size is 1024. We use the Adam optimizer [45] with a learning rate of $10^{-3}$.

**Downstream supervised classification:** For the linear and MLP classifiers, we use the cross-entropy loss to train the model. For KNN, we use the scikit-learn implementation [46] with the number of nearest neighbors $k = 5$.

## 4.4 Baselines

**LOLCAT and its variants:** LOLCAT [31] is a supervised framework for predicting cell types from individual neuronal activities using a multi-head attention network. Since the attention in LOLCAT is a simple weighted sum operation, we implement two additional supervised variants Transformer+ISI and Transformer+Raw using the self-attention mechanism employed in NeuPRINT for a fair comparison. These two variants follow the attention design of [28] and use a special classification token to represent the classification of the entire neuronal activity [47]. Transformer+ISI operates on the inter-spike interval (ISI) distributions input summarized from non-overlapping sub-windows of continuous 2-photon calcium recordings. We use suite2p package [48] to infer spikes from raw calcium traces and compute the ISI distributions. On the other hand, Transformer+Raw operates directly on the raw calcium traces. Unlike NeuPRINT, LOLCAT and the two supervised variants train both the attention network and classifier (linear or MLP) in an end-to-end fashion, using neuronal class and subclass labels during learning (See Appendix for details). As a consequence of the supervised training scheme, LOLCAT does not extract time-invariant representations of neuronal identity and its performance is constrained by the number of labels available in the dataset.

**Principal component analysis:** We project the raw calcium activities to a low-dimensional representation using Principal Component Analysis (PCA) and evaluate the effectiveness of this representation for downstream tasks. The projection is performed on a randomly selected sub-window in the raw recordings of each neuron, therefore its projected representation is also not time-invariant.

**Uniform manifold approximation and projection:** Similar to PCA, we project the raw calcium activities to a low-dimensional representation using Uniform Manifold Approximation and Projection (UMAP) [49], and evaluate the resulted time-variant representation by downstream classifiers.

**Random:** We further generate random representations for individual neurons, and train a supervised classifier on the random representations to measure the chance level of prediction.

## 5 Results

### 5.1 Self-supervised Learning Demonstrates Superior Generalization Capabilities in Data-Limited Scenarios

We evaluate our proposed method NeuPRINT and other baselines under three categories as shown in Table 1: (i) time-invariant vs. time-variant; (ii) self-supervised representation learning vs. end-to-end

| Input | — | | | | | | | Individual without Population | | | | Individual with Population | | | |
|---|---|---|---|---|---|---|---|---|---|---|---|---|---|---|---|
| Supervision | Lower Bound | Data-Limited Supervised | | | Unsupervised | | Self-supervised | | Data-Limited Supervised | | | Self-supervised | | | |
| Task | Model | Random | LOLCAT | Trans +ISI | Trans +Raw | PCA | UMAP | NeuPRINT | LOLCAT | Trans +ISI | Trans +Raw | NeuPRINT | | | |
| Subclass | KNN | 0.260 | — | — | — | 0.263 | 0.281 | 0.415 | — | — | — | **0.610** | | | |
| | Linear | 0.256 | 0.404 | 0.474 | 0.474 | 0.316 | 0.404 | 0.537 | 0.474 | 0.491 | 0.386 | **0.683** | | | |
| | MLP | 0.302 | — | 0.561 | 0.439 | 0.330 | 0.340 | 0.512 | — | 0.526 | 0.386 | **0.756** | | | |
| Class | KNN | 0.488 | — | — | — | 0.536 | 0.584 | 0.652 | — | — | — | **0.711** | | | |
| | Linear | 0.526 | 0.600 | 0.664 | 0.669 | 0.544 | 0.576 | 0.697 | 0.608 | 0.680 | 0.608 | **0.793** | | | |
| | MLP | 0.523 | — | 0.640 | 0.664 | 0.565 | 0.520 | 0.752 | — | 0.632 | 0.616 | **0.807** | | | |

Table 1: Top-1 accuracy of transcriptomic label prediction based on representations learned by (i) our proposed self-supervised representation learning from neural dynamics framework NeuPRINT, (ii) the supervised learning method LOLCAT [31] and its variants Transformer+ISI and Transformer+Raw, (iii) unsupervised baselines PCA and UMAP, (iv) random representations (to determine the chance-level). Note that this experiment corresponds to a data-limited regime due to limited labeled data. We performed classification using three classifiers (KNN, Linear, MLP) and two tasks: predicting the subclass from the set {Lamp5, Pvalb, Vip, Sncg, Sst}, and predicting the cell class from the set {excitatory, inhibitory}. We study the performance of different models using only individual neuronal activity vs adding population statistics as input.

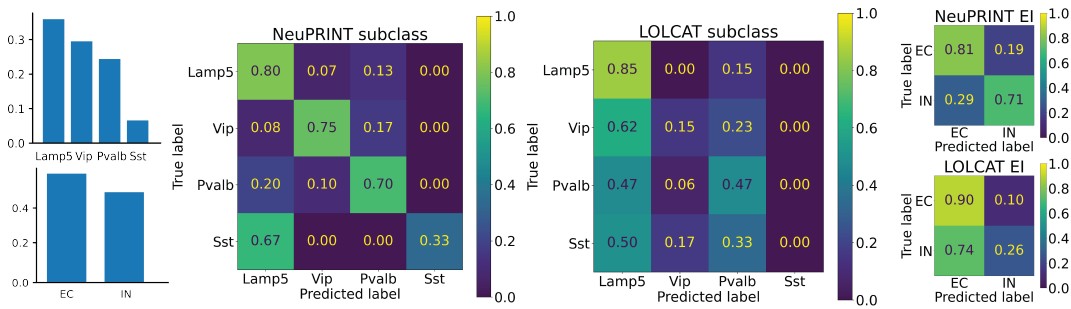

Figure 2: **Left:** Relative abundances of subclass and class labels. **Right:** Confusion matrices of our self-supervised representation learning framework NeuPRINT and supervised learning method LOLCAT [31] based on predicting the cell class and subclass labels. While both of the self-supervised and supervised steps are learned with all available subclasses, we excluded the Sncg population from the confusion matrices because it represents a negligible fraction of the test set with the $80\% : 10\% : 10\%$ split, so that quantification for this population would not be reliable.

supervised learning vs. unsupervised learning; (iii) activity of the neuron of interest ("Individual") as the only input to the dynamics model $f$ vs. permutation-invariant representation of population dynamics provided as exogenous input to $f$. Under the data-limited scenario (i.e., a small amount of labeled samples), which describes a vast majority of neuroscience datasets, we find that our self-supervised representation learning model NeuPRINT with a supervised downstream MLP classifier outperforms the current state-of-the-art approach LOLCAT by $> 35\%$ and its variants by $> 19\%$ in the subclass prediction task and outperforms LOLCAT by $> 20\%$ and its variants by $> 13\%$ in the class prediction task. Since LOLCAT is optimized using features extracted from non-overlapping sub-windows in an end-to-end supervised learning approach, it does not generate time-invariant representations that are critical to generalize across trials. Moreover, as shown in the confusion matrices in Figure 2, when the data in the subclass and class prediction tasks has an imbalanced distribution under the data-limited regime, our method can generate more balanced predictions across labels than LOLCAT. Both of these methods outperform two other standard unsupervised representation learning baselines, PCA and UMAP, which also extract time-varying representations across non-overlapping sub-windows. All of the evaluated models perform above the chance level, and we find that one-hidden layer MLP classifier improves classification accuracy over the linear classifier or KNN.

Ablation studies (Table 1 and Figure 3) show that using a permutation-invariant summary of population dynamics is critical to improving NeuPRINT's performance on the downstream subclass prediction.

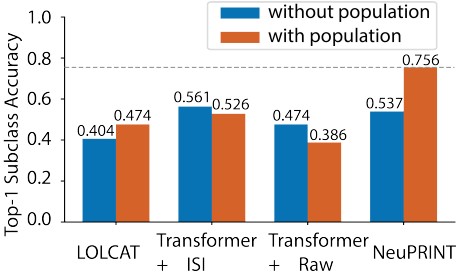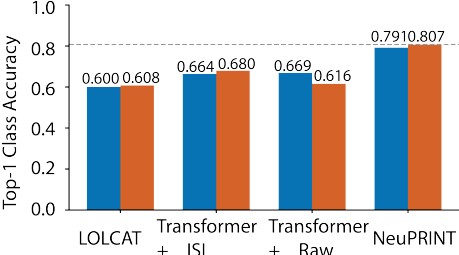

Figure 3: Accuracy of transcriptomic subclass and class prediction of NeuPRINT and baselines on single-mouse spontaneous activity recordings, with and without inputs from population statistics.

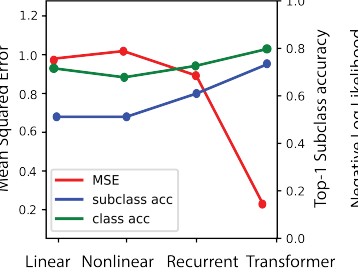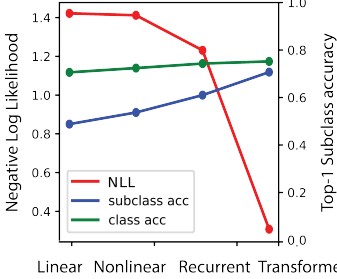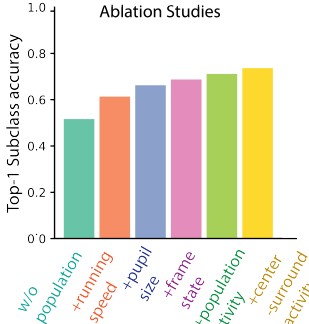

Figure 4: **Left:** Top-1 accuracy (or loss) of learned representations with different dynamical models (linear, nonlinear, recurrent, transformer) in the subclass prediction task. **Right:** Ablation studies to dissect the impact of the different components of the permutation-invariant summary of population dynamics including running speed, pupil size, frame state, population activity, center-surround activity in improving the accuracy, as in Table 5. One component is added at a time from left to right.

On the other hand, this effect is not significant in the class prediction task, suggesting that the intrinsic electrophysiology of neurons is significantly different between the excitatory vs inhibitory classes.

## 5.2 Learning Representations Across a Spectrum of Implicit Dynamical Models

We next investigate learning representations using our NeuPRINT framework over a spectrum of implicit dynamical models ranging from simple linear and nonlinear dynamical models to more advanced deep learning architectures such as gated recurrent networks and transformers. We also evaluate the performance of the models under two different objective functions (mean squared error vs Gaussian negative log likelihood). The results are shown in Figure 4. We find that the transformer, which leverages the powerful attention mechanism [28] to preserve the information over a large temporal context, achieves the best performance in predicting the masked (future) neural activities, as quantified by NLL and MSE loss. This ability to capture the dynamics more faithfully explains the transformer's superior performance in inferring neuron identity since neuronal physiology correlates with molecular expression [10, 29]. We note that, for this relatively small dataset, the MSE loss performs better than the NLL loss based on the downstream classification accuracy.

## 5.3 Permutation-Invariant Summary of Population Dynamics Enhances the Time-invariant Representation of Individual Neurons

To further investigate the role of each component in the permutation-invariant summary of population dynamics, we perform a series of ablation studies as shown in Figure 4. We add each input (running speed, pupil size, frame state, population activity, and center-surround activity) to NeuPRINT one at a time. We find that all of the proposed components contribute to the success of the time-invariant representations as evaluated by the downstream transcriptomic classification task. These results support the perspective put forth in Section 3: how the neuron reacts to external inputs (hence the summary variables proposed here) forms a part of the neuron identity. Overall, we find using all of the available permutation-invariant components of the summary of the population recording improves the accuracy by 22% in subclass prediction.

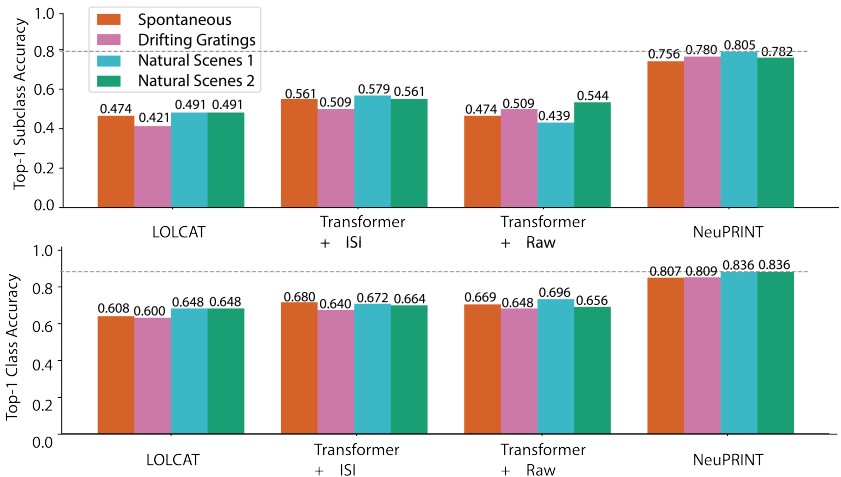

Figure 5: Accuracy of transcriptomic subclass and class prediction of NeuPRINT and baselines on single-mouse recordings during spontaneous activity and visual stimuli-driven (drifting gratings, natural scenes) activity.

| Task | Learned Representation Classifier | Lower Bound Random | Data-Limited Supervised LOLCAT | Trans +ISI | Trans +Raw | Unsupervised PCA | UMAP | Self-supervised NeuPRINT without population | Self-supervised NeuPRINT (ours) |
|---|---|---|---|---|---|---|---|---|---|
| | Inputs | — | Individual | | | | | | Individual + Population |
| Subclass | KNN | 0.174 | — | — | — | 0.333 | 0.348 | 0.419 | **0.552** |
| | Linear | 0.340 | 0.457 | 0.449 | 0.493 | 0.304 | 0.384 | 0.552 | **0.590** |
| | MLP | 0.362 | — | 0.442 | 0.423 | 0.384 | 0.406 | 0.552 | **0.685** |
| Class | KNN | 0.581 | — | — | — | 0.670 | 0.613 | 0.659 | **0.700** |
| | Linear | 0.667 | 0.700 | 0.710 | 0.675 | 0.645 | 0.660 | **0.746** | **0.746** |
| | MLP | 0.660 | — | 0.702 | 0.707 | 0.682 | 0.667 | 0.770 | **0.800** |

Table 2: **Extensions to multiple animals**: Top-1 accuracy of transcriptomic label prediction based on (i) the representations learned by our proposed self-supervised representation learning from neural dynamics framework NeuPRINT, (ii) the supervised learning method LOLCAT and its variants Transformer+ISI and Transformer+Raw, (iii) unsupervised baselines PCA and UMAP, (iv) random representations (to determine the chance-level). Note that this experiment corresponds to a data-limited regime due to limited labeled data. We performed classification using three classifiers (KNN, Linear, MLP) and two tasks: predicting the cell class from the set {excitatory, inhibitory} and predicting the subclass from the set {Lamp5, Pvalb, Vip, Sncg, Sst}.

## 5.4 Cell Type Identifiability Tends to Increase with Stimulus Relevance and Complexity

We further test our model NeuPRINT and other baselines on recordings with 3 sets of visual stimuli (drifting gratings and two different natural scene image sets [6]). The results summarized in Figure 5 suggest an increase in cell type identification accuracy of NeuPRINT from *in-vivo* activity as stimulus relevance and complexity increases, e.g., higher accuracy ($\sim 5\%$) in subclass prediction based on Natural Scenes 1 compared to spontaneous activity recording.

## 5.5 Extensions to Multiple Animals

We further extend our evaluations from one animal to multiple animals. We report the evaluations for all animals on the extended dataset in Table 2. While the performance of LOLCAT increases on the class prediction task with this larger dataset, it still remains less accurate than our model across all tasks and classifiers. For subclass prediction, where the distribution of cells across subclasses is highly imbalanced, our method NeuPRINT outperforms LOLCAT by $\sim 23\%$, and also by $\sim 10\%$ in class prediction (See discussion on the preliminary nature of this study below).

# 6  Discussion and Conclusion

In this work, we studied the problem of inferring neuronal identity from *in vivo* recordings. Motivated by the fact that *in vivo* physiology of a neuron has two distinct components (the neuron itself, and the synaptic and modulatory inputs it receives), we presented a self-supervised approach to infer identity vectors for neurons based on a model of neuronal dynamics with exogenous inputs. While letting the activities of neighboring neurons represent the exogenous input directly could be a natural choice, this would prohibit the transferability of the model because a biologically meaningful ordering of neighboring neurons is not available in these recordings. This key observation led us to propose calculating permutation-invariant statistics of the activity of the neighboring neurons.

In the field of natural language processing (NLP), learning the dynamics of the underlying system has been shown to enable success in multiple downstream tasks [43]. Inspired by this, we proposed to use the identity vectors inferred by our dynamics model in predicting the molecular labels of the neurons. This was enabled by a recent, public dataset that profiles both *in vivo* calcium activity and subsequent transcriptomic expression in individual neurons [6]. Learning the dynamics is a self-supervised task. We trained simple classifiers with limited ground-truth labels to subsequently predict the molecular label, based on the invariant representations inferred by this self-supervised step. We believe this two-step approach could constitute a benchmark for future studies on this topic.

We experimented with different dynamics models. We reported that, consistent with the impressive findings in NLP [28, 43], the transformer architecture with its self-attention mechanism provides the best results among the architectures we considered. Due to its feedforward structure, this architecture also enables fast inference, which could be useful for experimental setups with real-time feedback.

Ablation experiments demonstrated the merit of providing permutation-invariant inputs and information on the overall state of the nervous system (e.g., pupil size). This set of experiments also revealed that more detailed input representations could further improve performance while retaining the desired permutation-invariance property. In particular, motivated by insights on circuit motifs in neuronal networks [12–14], we demonstrated a center-surround setup, which partitions the available population into two concentric sets and calculates identical statistics for each set separately.

Our approach can be improved when applied to multi-region, cross-session and cross-animal recordings by learning extra embeddings that represent different brain regions, sessions or animals. In this sense, our multi-animal experiments represent a limited, preliminary study. It may also be possible to study recordings from healthy vs diseased brains or from individuals belonging to different species using a single model.

Our study represents an initial attempt. Ever-growing neuroscience datasets represent a straightforward way to improve accuracy. On the technical side, engineering of the invariant features can extend the list of meaningful and invariant statistics of population activity. Contrastive learning of neuronal identity can improve classification accuracy by encouraging the model to depend more on the identity input. Our center-surround setup should also be considered as an initial attempt at capturing existing neurobiological insights in the kind of dynamics models studied in this paper. Studying the transferability of the proposed model across individuals is a high-priority direction for future research.

# 7  Reproducibility

All optimizations are performed on one NVIDIA Tesla V100 GPU. Details are included in Appendix E. We released our software (`https://github.com/lumimim/NeuPRINT/`) for reproducibility.

# 8  Acknowledgement

LM is supported by the Shanahan Foundation Fellowship. LM and US thank the Allen Institute founder, Paul G Allen, for his vision, encouragement, and support. ES and TL acknowledge the support in part by A3D3 National Science Foundation grant OAC-2117997 and the Department of Electrical Computer Engineering at the University of Washington. ES also acknowledges the support in part by the Washington Research Foundation Fund and the Department of Applied Mathematics at the University of Washington.

**Broader Impact**    Since we studied a basic neuroscience problem using recordings from mice, we do not expect this paper to have an immediate societal impact. We hope that, in the long term, variants of our method will be helpful to better understand how the brain works and the physiological consequences of the various diseases of the brain. It is nevertheless important to keep in mind that technical progress, such as reported in our study, could potentially be adapted without much difficulty to malicious use.

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

| Task | Learned Representation | Time-Variant | | | | | Time-Invariant | |
|---|---|---|---|---|---|---|---|---|
| | | Lower Bound | Data-Limited Supervised | | Unsupervised | | Self-supervised | |
| | Inputs | — | Individual | | | | | Individual + Population |
| Task | Classifier | Random | LOLCAT | Trans +ISI | Trans +Raw | PCA | UMAP | NeuPRINT without population | NeuPRINT (ours) |
| Subclass | KNN | 0.196 ± 0.041 | — | — | — | 0.345 ± 0.047 | 0.371 ± 0.070 | 0.424 ± 0.051 | **0.546 ± 0.056** |
| | Linear | 0.326 ± 0.055 | 0.449 ± 0.027 | 0.414 ± 0.036 | 0.436 ± 0.021 | 0.316 ± 0.046 | 0.386 ± 0.046 | 0.595 ± 0.041 | **0.683 ± 0.042** |
| | MLP | 0.382 ± 0.062 | — | 0.456 ± 0.068 | 0.422 ± 0.017 | 0.392 ± 0.044 | 0.496 ± 0.054 | 0.606 ± 0.054 | **0.722 ± 0.022** |
| Class | KNN | 0.509 ± 0.061 | — | — | — | 0.569 ± 0.015 | 0.547 ± 0.046 | 0.568 ± 0.057 | **0.705 ± 0.017** |
| | Linear | 0.536 ± 0.035 | 0.616 ±0.017 | 0.632 ± 0.026 | 0.652 ± 0.009 | 0.584 ± 0.053 | 0.583 ± 0.019 | 0.741 ± 0.035 | **0.766 ± 0.031** |
| | MLP | 0.565 ± 0.030 | — | 0.624 ± 0.010 | 0.651 ± 0.007 | 0.596 ± 0.015 | 0.584 ± 0.013 | 0.782 ± 0.018 | **0.803 ± 0.037** |

Table 3: **Sensitivity analysis across 5 runs with different random seeds**: Top-1 accuracy (**mean±standard deviation**) of transcriptomic label prediction based on (i) the representations learned by our proposed self-supervised representation learning from neural dynamics framework NeuPRINT, (ii) the supervised learning method LOLCAT and its variants Transformer+ISI and Transformer+Raw (abbreviated Trans+ISI and Trans+Raw, respectively) (Note that this experiment corresponds to a data-limited regime due to the size of the dataset), (iii) unsupervised baselines PCA and UMAP, (iv) random representations (to determine the chance-level). We performed classification using three different simple classifiers (KNN, Linear, MLP) and two tasks: predicting the cell class from the set {excitatory, inhibitory} and predicting the subclass from the set {Lamp5, Pvalb, Vip, Sncg, Sst}.

# A  Sensitivity Analysis

To quantify the sensitivity of our methods and baselines on the transcriptomic identity (neuron class and subclass) prediction benchmark, we run each model with 5 different random seeds (train/val/test random split, random initializations, etc.). We report their averaged Top-1 accuracies and the corresponding standard deviations across 5 runs for all models in Table 3.

Consistent with the main text, we find NeuPRINT still significantly outperforms other methods including supervised baselines LOLCAT [31] and its variants Transformer+ISI and Transformer+Raw (Note the limited availability of labeled data). The standard deviations for all methods are relatively small, indicating the robustness of our evaluation framework and the significance of the performance gaps.

# B  A Spectrum of Implicit Dynamical Models

We explore a spectrum of implicit dynamical models – Linear, Nonlinear, Recurrent neural network, Transformer, and two objective functions – mean squared error (MSE) when the model only predicts the mean of the output distribution; negative log likelihood (NLL) when the model predicts both mean and standard deviation of output distribution to train the models. The results are summarized in Table 4.

# C  Roles of Permutation-Invariant Summary of Population Dynamics

To investigate the role of each component in the permutation-invariant summary of population dynamics, we perform a series of ablation studies as shown in Table 5. We add each input (running speed, pupil size, frame state, population activity, and center-surround activity) to NeuPRINT one at a time. We find that all of the proposed components contribute to the success of the time-invariant representations as evaluated by the downstream transcriptomic classification task.

| Task | Linear | | Nonlinear | | Recurrent | | Transformer | |
|---|---|---|---|---|---|---|---|---|
| Objective | NLL | MSE | NLL | MSE | NLL | MSE | NLL | MSE |
| Loss | 1.422 | 0.984 | 1.412 | 1.022 | 1.231 | 0.887 | 0.308 | **0.221** |
| Subclass | 0.488 | 0.512 | 0.537 | 0.512 | 0.610 | 0.610 | 0.707 | **0.756** |
| Class | 0.706 | 0.716 | 0.724 | 0.679 | 0.743 | 0.743 | 0.752 | **0.807** |

Table 4: Comparison of reconstruction loss (first row, the smaller the better) and top-1 accuracy of class and subclass prediction using MLP downstream classifier (second and third rows, the larger the better) achieved by different dynamics models $f$ (linear, nonlinear, RNN, and transformer) in NeuPRINT. Two objectives are used to train each model (mean squared error (MSE) or negative log likelihood (NLL)).

| Subclass | w/o population inputs | + running speed | + pupil size | + frame state | + population activity | + center-surround activity |
|---|---|---|---|---|---|---|
| MLP | 0.512 | 0.609 | 0.658 | 0.683 | 0.732 | **0.756** |

Table 5: Ablation studies of permutation-invariant inputs representing population activity, including running speed, pupil size, frame state, permutation-invariant population representation, permutation-invariant center-surround representation. Results are reported on the subclass prediction task with an MLP downstream classifier.

# D  Extensions to Other Visual Stimulus Conditions

We further test our model NeuPRINT and other baselines on recordings with 3 sets of visual stimuli (drifting gratings and two different natural scenes). The results in Table 6 suggest an increase in cell type identification accuracy from in-vivo activity as stimulus relevance and complexity increases.

# E  Implementation Details

## E.1  NeuPRINT

**Permutation-invariant summary of population dynamics:** We use $C$, $S$ to represent $D$, $D$ dimensional permutation- and size-invariant (i.e., $N$) surrogates for brain activity at each time point. $C$ and $S$ compute identical statistics (mean and standard deviation) of population activity except that the statistics in $C$ are calculated over the center partition (neurons whose distance to neuron $i$ is at most $\Delta$) and the statistics in $S$ are calculated over the surround partition (neurons whose distance to neuron $i$ is larger than $\Delta$). Here, $\Delta$ is a hyperparameter:

$$\mu_{C_t^{(-i)}} = \text{mean}(X_j(t) \,|\, j : 0 < d_{ji} < \Delta) \tag{4}$$

$$\sigma_{C_t^{(-i)}} = \text{std}(X_j(t) \,|\, j : 0 < d_{ji} < \Delta) \tag{5}$$

$$\mu_{S_t^{(-i)}} = \text{mean}(X_j(t) \,|\, j : 0 < \Delta \le d_{ji}) \tag{6}$$

$$\sigma_{S_t^{(-i)}} = \text{std}(X_j(t) \,|\, j : 0 < \Delta \le d_{ji}) \tag{7}$$

**Batch sampling:** For each batch of data we randomly sample 512 labeled and unlabeled neurons to be included in the batch. For each neuron we further randomly sample 2 sub-windows of 512 timesteps from its continuous calcium flourescene traces. Each resulting sample for the $i^{th}$ neuron is denoted $X^{(i)}$.

**Multihead attention:** We use the transformer encoder architecture [28] and the masking strategy as originally proposed in [47]. Random timesteps in $X^{(i)}$ are masked (zero-out) with probability 0.25 and concatenated with the permutation-invariant population summary $\bar{P}$ and time-invariant representation $\phi^{(i)}$ along the feature dimension to form input $\bar{X}^{(i)}$. Note that the same learnable $\phi^{(i)}$ is repeated at every timestep, enforcing time-invariance. Similar to [47], we embed input $\bar{X}^{(i)}$ and

| | Task | Classifier | LOLCAT | Transformer+ISI | Transformer+Raw | NeuPRINT |
|---|---|---|---|---|---|---|
| Spontaneous Activity | Subclass | linear | 0.474 | 0.491 | 0.474 | **0.683** |
| | | mlp | —- | 0.561 | 0.439 | **0.756** |
| | Class | linear | 0.608 | 0.680 | 0.669 | **0.793** |
| | | mlp | —- | 0.640 | 0.664 | **0.807** |
| Drifting Gratings | Subclass | linear | 0.421 | 0.509 | 0.404 | **0.756** |
| | | mlp | —- | 0.474 | 0.509 | **0.780** |
| | Class | linear | 0.600 | 0.640 | 0.648 | **0.809** |
| | | mlp | —- | 0.632 | 0.640 | **0.809** |
| Natural Scenes 1 | Subclass | linear | 0.491 | 0.579 | 0.421 | **0.780** |
| | | mlp | —- | 0.544 | 0.439 | **0.805** |
| | Class | linear | 0.648 | 0.672 | 0.696 | **0.809** |
| | | mlp | —- | 0.664 | 0.640 | **0.836** |
| Natural Scenes 2 | Subclass | linear | 0.491 | 0.509 | 0.544 | **0.732** |
| | | mlp | —- | 0.561 | 0.526 | **0.782** |
| | Class | linear | 0.648 | 0.664 | 0.656 | **0.809** |
| | | mlp | —- | 0.664 | 0.648 | **0.836** |

Table 6: Accuracy of transcriptomic subclass and class prediction of NeuPRINT and baselines on single-mouse recordings during spontaneous activity and visual stimuli-driven (drifting gratings, natural scenes) activity.

employ sinusoidal positional embedding to encode the temporal order in the input sequence, resulting in $\tilde{X}^{(i)} = \text{Emb}(\bar{X}^{(i)}) + \text{E}$.

For each input $\tilde{X}^{(i)}$, a set of weights $W^Q \in \mathbb{R}^{T \times d_q}$, $W^K \in \mathbb{R}^{T \times d_k}$, $W^V \in \mathbb{R}^{T \times d_v}$ are learned to transform input $\tilde{X}^{(i)}$ to a set of query, key, and value $(Q, K, V)$, where $Q = \tilde{X}^{(i)}W^Q$, $K = \tilde{X}^{(i)}W^K$, $V = \tilde{X}^{(i)}W^V$. Attention between temporal tokens for one attention head is computed as:

$$\text{Attention}(Q, K, V) = \text{softmax}\left(\frac{QK^\top}{\sqrt{d_k}}\right)V \tag{8}$$

Each head will find a different pattern in the data and produce an output of size $d_v$. The final attention output will be a concatenation of these single-head outputs. We use 2 heads in our model.

Feedforward layers and residual connections are subsequently applied to attention output:

$$Z^{(i)} = \tilde{X}^{(i)} + \text{MSA}(\tilde{X}^{(i)}) + \text{FF}(\tilde{X}^{(i)} + \text{MSA}(\tilde{X}^{(i)})) \tag{9}$$

where MSA represents the multihead attention operation, FF represents the feedforward layer with ReLU activation, and $Z^{(i)}$ represents the reconstructed calcium trace with masked timesteps recovered.

**Computational cost:** NeuPRINT has 833K parameters and takes 2 hours in total for training and inference on a single NVIDIA Tesla V100 GPU. Details on the size of the datasets are mentioned in Section 4.1 of the main paper.

### E.2 LOLCAT and Its Variants

We use the publicly available implementation of LOLCAT from [31]. We implement two additional variants of LOLCAT (Transformer+ISI and Transformer+Raw), following the philosophy in [31] but with the self-attention mechanism as used by NeuPRINT [28]. For LOLCAT and Transformer+ISI, continuous time series of calcium traces are divided into non-overlapping sub-windows of size 64, within which the Inter-Spike Interval (ISI) distribution is computed. We use suite2p Python package [48] to infer spikes and compute the ISI distribution with 16 bins, using a spike threshold of 0.2. To perform classification using the self-attention mechanism, we use a learnable special classification token [CLS] appended to the beginning of the input sequence to represent the classification output [47]. No positional embedding is added to the input sequence. Therefore the transformer output at the CLS token position will represent the pooling operation as in [31], and all sub-windows are treated as if they are independent trials, which enables Transformer+ISI to apply to any number of observed trials - an important design choice for LOLCAT.

We further implement Transformer+Raw, a variant of LOLCAT where the transformer operates directly on the raw calcium traces rather than the ISI distribution of calcium traces. Transformer+Raw

| | NeuPRINT | LOLCAT | Transformer +ISI | Transformer +Raw | random/ PCA/UMAP |
|---|---|---|---|---|---|
| Representation dim | 64 | — | — | — | 64 |
| Encoder hidden dim | 70 | [32, 16, 16] | $[128] \times 4$ | $[128] \times 4$ | — |
| MLP classifier hidden dim | 2048 | — | 2048 | 2048 | 2048 |
| Number of attention heads | 2 | 4 | 4 | 4 | — |
| Number of attention layers | 1 | 1 | 1 | 1 | — |
| Window size | 200 | 2048 | 512 | 512 | 2048 |
| Context window size | 2 | — | 8 | 1 | — |
| Batch size | 1024 | varies | varies | varies | — |
| Number of epochs | 400 | 500 | 500 | 500 | — |
| Number of downstream epochs | 5000 | — | — | — | 1000 |
| Learning rate | $10^{-3}$ | — | — | — | — |
| Downstream learning rate | $10^{-4}$ | $10^{-4}$ | $10^{-4}$ | $10^{-4}$ | $10^{-3}$ |
| Dropout | 0.1 | — | 0.1 | 0.1 | — |
| KNN neighbors | 5 | — | 5 | 5 | 5 |
| ISI sub-window size | — | 64 | 64 | — | — |
| Number of ISI bins | — | 16 | 16 | — | — |

Table 7: **Hyperparameters** of our self-supervised representation framework NeuPRINT and other baselines including end-to-end supervised learning model LOLCAT, its variants Transformer+ISI and Transformer+Raw, unsupervised representation learning models PCA and UMAP, and chance-level prediction based on random features.

follows the same architecture as Transformer+ISI, except that now the positional embedding is added to the input sequence to denote the temporal relationship between timesteps in the trial window.

### E.3 Random / PCA / UMAP

To probe the chance-level classification performance, we train and evaluate downstream classifiers using random vectors of size 64 as representations for individual neurons, equivalent to the 64-dim $\phi^{(i)}$. To compare NeuPRINT with unsupervised methods PCA and UMAP, we first project the data to a lower dimensional space using 64 components, then train and evaluate downstream classifiers on the low-dimensional representation.

### E.4 Downstream Classifiers

We use 5 nearest neighbors for KNN downstream classifier. For the downstream MLP classifier, we use a multi-layer perceptron network with a single hidden layer of size 2048 and ReLU activation.

### E.5 Hyperparameters

We include all of the important hyperparameters (representation dim, window size, number of epochs, learning rate, batch size, etc.) for our NeuPRINT and other models (supervised-learning baselines LOLCAT, Transformer+ISI and Transformer+Raw, unsupervised representation learning baselines UMAP and PCA, chance-level prediction based on random features) in Table 7.

## F Pseudo Code

Our NeuPRINT framework includes three main components: an implicit dynamical system that uses the state-of-the-art transformer architecture to model neural dynamics; an optimization framework that fits the dynamical model and learns time-invariant representations for neurons; a supervised learning framework to train the downstream classifiers for subclass and class prediction, taking the learned time-invariant representations as inputs. The pseudo code for these three components is listed as follows:

```
Transformer ( calcium_fluoroscence_trace ,
              permutation_invariant_population_summary ,
              time_invariant_representation ):

        input_embedder = linear ( input_dim , hidden_dim )

        transformer_encoder = multihead_attention (
                                          window_size ,
                                          layer_dim ,
                                          num_heads )

        output_decoder = linear ( hidden_dim , output_dim )

        masked_calcium_fluoroscence_trace =
                    mask_inputs ( calcium_fluoroscence_trace )

        input = concatenate (
                masked_calcium_fluoroscence_trace ,
                permutation_invariant_population_summary ,
                time_invariant_representation )

        input = positional_encoding ( input )

        input = input_embedder ( input )

        context_mask = generate_context_mask (
                                    window_size ,
                                    context_window_size )

        output = transformer_encoder ( input , context_mask )

        output = output_decoder ( output )

        return output

Time_Invariant_Self_Supervised_Representation_Learning (
                            calcium_fluoroscence_trace ,
                            permutation_invariant_population_summary ):

        time_invariant_representation = zeros ( neuron_dim , embedding_dim )

        recon_model = Transformer (
                            input_dim ,
                            hidden_dim ,
                            window_size ,
                            context_window_size ,
                            layer_dim ,
                            num_heads )

        optimizer = Adam (
                time_invariant_representation ,
                recon_model ,
                learning_rate )

        predicted_calcium_fluoroscence_trace = recon_model (
                calcium_fluoroscence_trace ,
                permutation_invariant_population_summary ,
                time_invariant_representation )

        recon_loss = mse (
                predicted_calcium_fluoroscence_trace [ masked_steps ] ,
                calcium_fluoroscence_trace [ masked_steps ])

        recon_loss . backward ()
```

```
        optimizer.step()

        return time_invariant_representation

Downstream_Classifier_Supervised_Learning(
                        time_invariant_representation,
                        ground_truth_class_label):

        classifier = mlp(embedding_dim, hidden_dim, output_dim)

        optimizer = Adam(classifier, learning_rate)

        predicted_class_label = classifier(time_invariant_representation)

        classification_loss = cross_entropy(
                        predicted_class_label,
                        ground_truth_class_label)

        classification_loss.backward()

        optimizer.step()

        return predicted_class_label
```

