# OpenReview forum: "Learning Time-Invariant Representations for Individual Neurons from Population Dynamics"
_NeurIPS.cc/2023/Conference — NeurIPS 2023 poster_

### Official Review · Reviewer_4tWo · 2023-06-20

**Soundness:** 3 good
**Presentation:** 3 good
**Contribution:** 2 fair
**Rating:** 5
**Confidence:** 3

**Summary:**

This paper introduces NeuPRINT, an unsupervised approach that simultaneously learn (i )a dynamical model of recorded neural activity and (ii) a time invariant embedding of individual neurons using transformers.

Using a downstream transcriptomic classification task, the authors show that the learnt embedding space contains relevant information about the neurons identity, and demonstrated improved performances against a range of baseline methods.

**Strengths:**

The proposed method not only allows to model the dynamical systems governing individual neurons recorded activity, but it also links stereotyped temporal motifs to time invariant location in an embedding space. The latter can therefore be interpreted as a representation of the intrinsic properties of the recorded neurons, like their conductance or capacitance.

The approach is conceptually interesting and could have interesting application in characterizing neural population from temporal traces. The authors designed an original model that uses summary of population activity as input to each neurons.

The approach is adapted on a range of architecture, and demonstrates that one can leverage the flexibility of transformers to find embedding more informative about neural identity than with traditional model like linear dynamical system.


**Weaknesses:**

The method description is not detailed enough and does not allow to reproduce the results presented in the paper. Overall, it is not very well written.

1) The statistics used to build $C_t$ and $S_t$ are never defined. Do they correspond to 1st and 2nd moment of calcium traces or to more complicated statistics ?

2) Only pseudo-code is provided.

3) Almost half a page is dedicated to Hodgkin-Huxley equations, which have no technical value in the present paper. Surely it would be possible to link the time invariant neural embedding with intrinsic properties of the neurons (like its shape, conductance, capacitance etc.) without introducing many variables that are never used subsequently. In contrast, very little is said about the new method itself.

4) Section 4.2:  without further explication, one cannot understand the difference between the two losses used. "Negative Log Likelihood of Gaussian"  (NLE) and Mean Squared Error (MSE) are exactly the same thing if the variance of the former model is 1. If the variance is not modeled/learned, then they only differ from multiplicative and additive constant. The fact that the authors did not find a monotonic relationship between MSE and NLE in Table 2 could mean two things: that (i) they indeed used different loses (in which case the method description should be more detailed), or (ii) that the same loss was used in both columns and that the difference in performances simply results from the use of different random seeds.

5) Section 1.1 "Related work" needs rewriting. NeuPRINT is interesting in that it combines several feature of well known neural data analysis techniques. For example, the use of summary statistics of population activity $P_t$ echoes many latent models (see Saxena 2019). Yet, contrary to standard techniques like GPFA, the influence of the population on individual is captured by equation (3), which relates the NeuPRINT to coupled model like (Pillow 2008 or Truccolo 2004). In its current state, section 1.1 simply list existing approaches without really relating them to proposed approach.

Minor Points:

1) The vocabulary is often vague (*e.g.* "numerous insights into how the brain works", line 20) and many abreviations like "w/" or "w/o" are unecessarily used in the main text.

2) I agree that the term "representation" is conveniently overloaded but the paper would gain in clarity by attempting to better explicit it. In the present study, from my understanding, the representation of a neuron is an "embedding of its time-invariant properties that have an effect on its dynamics".

**Questions:**



1) How are $C$, $S$ precisely defined ? (at least in the experiment)

2)  The author only evaluate their model using the discovered embedding space. What about the dynamics $f$ ? How do the calcium traces reconstructed by the model compare with linear dynamical systems for example ?

3) Why is the dependence on $i$ dropped from equation (3) to equation (4) ? Although $P$ must be permutation invariant, it might be relevant to model slightly different input $P^{-i}$ for each neuron. Since it does not seem much more computationally expendive to do so, can the author motivate this modeling choice ?

4) How is the embedding dimension chosen ? Can it be interpreted like other method like t-SNE or UMAP ?

5) Would it be possible to jointly learn $f$, $\Phi$ *and* $P_t$ ? In doing so, the model would yield interpretable embedding and neural trajectories !



**Limitations:**

The authors adequately addressed the ethical limitations of their paper.

---

> ### Author Rebuttal · Authors · 2023-08-10
>
> We thank the reviewer for the constructive comments on improving the writing and presentation of the paper. Meanwhile, we encourage the reviewer to also check the results in the overall response and attached pdf.
>
> **“The statistics used to build C  and S are never defined.”**
>
> We provide the precise definitions of $C_t$ and $S_t$ as below. The reviewer is right that $C^{(-i)}_t$ and $S^{(-i)}_t$ correspond to the 1st moment (mean) and 2nd moment (standard deviation) of calcium traces of the center and surrounding partitions of every neuron $i$. We will explicitly include this definition in our revision.
>
> $$\mu\_{C^{(-i)}\_t} = \text{mean}(X\_t^j | j: 0<d\_{ji} < \Delta)$$
> $$\sigma\_{C^{(-i)}\_t} = \text{std} (X\_t^j | j: 0<d\_{ji} < \Delta)$$
> $$\mu\_{S^{(-i)}\_t} = \text{mean}(X\_t^j | j: d\_{ji} \geq \Delta)$$
> $$\sigma\_{S^{(-i)}_t} = \text{std} (X\_t^j | j: d\_{ji} \geq \Delta)$$
>
> where $\text{mean}$ and $\text{std}$ calculate empirical mean and standard deviation of their argument, respectively, $d_{ji}$ denotes the distance between cell bodies of neurons $i$ and $j$, and $\Delta$ denotes the distance hyperparameter partitioning center vs surround.
>
> **“Only pseudo-code is provided.”**
>
> We will definitely provide the source code for reproducibility after the paper is published.
>
> **“Almost half a page is dedicated to Hodgkin-Huxley equations, which have no technical value in the present paper.”**
>
> We presented Hodgkin-Huxley equations to provide biological motivation on the need to introduce the time-invariant components that contribute to neuronal dynamics. As a phenomenological model, Hodgkin-Huxley equations give an explicit format of time-invariant components and the dynamics, while in our methods, both components are defined implicitly. We understand the reviewer’s concern that this introduces extra variables which might be confused as belonging to our main method. We will reframe and move the text surrounding the Hodgkin-Huxley equations to the Related Works section in our revised paper.
>
> **“The difference between the two losses used”**
>
> We would like to clarify that the two losses are in fact not the same. In our implementation, when trained with Gaussian Negative Log Likelihood loss (NLL), our model predicts a time-invariant Gaussian distribution (both mean and variance) for the trace at the predicted time step. Therefore the variance is learnable and is not simply set to 1. When trained with Mean Square Error loss (MSE), our model only predicts one output for the predicted time step, upon which the mean square error with the ground truth is calculated.
>
> **“Relating existing approaches to the proposed method”**
>
> We appreciate the reviewer’s suggestions on introducing the links between our paper and previous works. We will cite these references as suggested in the revised version.
>
> **“Definition of the term “representation”**
>
> We will explicitly define the term "representation" in our revised paper as "embedding of a neuron’s time-invariant properties that have an effect on its dynamics" as the reviewer correctly suggested.
>
> **“The authors only evaluate their model using the discovered embedding space. What about the dynamics? How do the calcium traces reconstructed by the model compare with linear dynamical systems for example?”**
>
> As shown in Table 2 and Figure 3 of the main paper how our method compares to other dynamical modeling approaches in terms of reconstructing the calcium traces. We evaluated the reconstruction loss via two metrics: NLE and MSE, which are shown numerically in Table 2 as “loss” row, and graphically in Figure 3 as red curves with left axis readings on both plots. We showed that the reconstruction loss for our transformer-based NeuPRINT method is significantly smaller (MSE: 0.221, NLE: 0.308) compared to other dynamical models, e.g. Linear (MSE: 0.984, NLE: 1.422), Recurrent (MSE:0.887, NLE:1.231), showcasing our transformer-based model’s ability to reconstruct neuronal dynamics with significantly better performance.
>
> **“Why is the dependence on i dropped from equation (3) to equation (4)? Although P must be permutation invariant, it might be relevant to model slightly different input $P^{-i}$ for each neuron?”**
>
> We thank the reviewer for catching the mismatch between equation 3 and equation 4. We will update our dynamical equation 4 in the main paper as follows:
>
> $$X^{(i)}\_{t+1} =  f(X^{(i)}\_{t-W+1:t}, (C^{(-i)}\_{t-W+1:t}, S^{(-i)}\_{t-W+1:t}, B\_{t-W+1:t}), \Phi^{(i)})$$
>
> where for each neuron i, the 1st and 2nd moment of activities of neighboring neurons in the “center” partition $C^{(-i)}$ and “surround” partition $S^{(-i)}$, as well as the global behavior statistics $B$, together contribute to neuron $I$’s dynamics as external input components. Note that B does not depend on $I$ as it represents global, behavioral statistics.
>
> **“How is the embedding dimension chosen? Can it be interpreted like other method like t-SNE or UMAP?”**
>
> The embedding dimension for our model is a hyperparameter that needs to be tuned to improve the reconstruction tasks and downstream tasks, it is similar to t-SNE or UMAP in the sense that it's for dimension reduction.
>
> **“Would it be possible to jointly learn $f$, $\Phi$ and $P_t$?”**
>
> We thank the reviewer for the constructive comments. Currently only time-invariant representation $\Phi$ and dynamic model $f$ are jointly learned, while $P$ is directly summarized by permutation-invariant operations with no learnable parameters, i.e. 1st and 2nd moments of population activity, since we would like the model to be invariant to varied size of neuronal population across different sessions. We agree that learning $P$ represents a valuable extension. We believe straightforward approaches will have difficulty in ensuring permutation invariance of $P$. Therefore, we did not immediately attempt to incorporate this idea. Nevertheless, we decided to mention this extension in the revised Discussion.

---

> > ### Comment · Reviewer_4tWo · 2023-08-14
> >
> > I thank the authors for their response and their helpful comments. I updated my score.
> > (Please note: my question #2 was irrelevant as the response was in the original draft. I think I forgot to remove it during my initial review)

---

> > > ### Author Response · Authors · 2023-08-15
> > >
> > > We thank the reviewer for considering our response and improving their score.

---

> > > ### Author Response · Authors · 2023-08-15
> > >
> > > Meanwhile, we were wondering if our response letter was able to address some of the reviewer's concerns regarding the soundness, presentation, contribution aspects. We were wondering if the reviewer would consider it appropriate to also update those scores to better reflect the reviewer's overall assessment of 'borderline accept'. Thank you.

---

> > > > ### Comment · Reviewer_4tWo · 2023-08-15
> > > >
> > > > Good Point. I updated 2 of them.

---

> > > > > ### Author Response · Authors · 2023-08-15
> > > > >
> > > > > We thank the reviewer for improving the scores accordingly.

---

### Official Review · Reviewer_MNFF · 2023-07-02

**Soundness:** 2 fair
**Presentation:** 2 fair
**Contribution:** 3 good
**Rating:** 5
**Confidence:** 4

**Summary:**

In this manuscript the authors describe an approach to predicting the transcriptomic identity of a neuron from its activity dynamics that they call NeuPRINT. Specifically, the authors take a dataset with calcium imaging and transcriptomic identification of the imaged neurons. First, they train up a model – either linear, nonlinear, recurrent, or a transformer – to predict the next timestep of the calcium trace using both past activity and what they term "time-invariant" parameters. Next, a downstream classifier is trained to use the learned parameters of the model to predict the genetic identity of each neuron. The authors demonstrate that their model outperforms the current state-of-the-art along with a series of control baselines.

**Strengths:**

Strengths:

1. Their model includes some interesting ideas. Namely, the use of population activity and time-invariant parameters seems to confer substantial predictability.
1. The performance of the model appears to be impressive on this multi-modal dataset.
1. Per the ablation study shown in Figure 3, the model appears to make use of all features that were fed to it.



**Weaknesses:**

Weakness.

Major:

1. A major claim made by the authors is that their method outperforms both LOLCAT and simple, effectively random baselines. While their model does appear to perform better, this does not look like an apples to apples comparison.
    1. First, it would be appear that LOLCAT uses *only* a neuron's activity in order to classify its transcriptomic identity. On the other hand, NeuPRINT includes activity from nearby neurons in addition to behavioral covariates such as pupil size and running speed. Unless I am mistaken, performance comparisons between NeuPRINT and LOLCAT with similar inputs (single neuron activity *only*) are not shown until the ablation study in Figure 3. Figure 3 right shows performance that appears to be similar to LOLCAT. Is the boost in performance with NeuPRINT solely coming from the population activity and behavioral covariates? If so, then authors would need to scale back their claims about superior performance. Alternatively, the authors could add this information to LOLCAT for a more fair comparison.
    1. Second, I am a little confused about the PCA/UMAP baselines. Here the authors selected a *random* window of time to project the data down to 64 components. Why 64? Did this account for a specific amount of variance? Why a random window of time? How was the window size chosen, and why not use a sliding window approach or use the full time-series?
1. Whether or not NeuPRINT outperforms LOLCAT, the LOLCAT paper systematically interrogates their model for insight into the underlying biology of neural systems. I could not find any comparable biological insight here.

Minor:

1. The discussion of time-variant and time-invariant is fairly confusing. For example, in line 233 in the description of LOLCAT, it is asserted that "As a consequence of the supervised training scheme, LOLCAT does not extract time-invariant representations of neuronal identity and its perfomance is constrained by the number of labels available in the dataset." Are the authors saying that ISI distributions are time-variant? Also, why wouldn't the performance of *any* classifier be limited to the available labels? I see no reason why this wouldn't also apply to NeuPRINT.
1. I found a number of typos, the manuscript needs a copy-edit.


**Questions:**

1. In Table 1 does the w/o population column mean NeuPRINT *w/o population or behavioral covariates*? In other words, does NeuPRINT have access to information that LOLCAT does not?
1. Can the authors interrogate the learned parameters of their model to learn anything new about the dataset they are modeling?

**Limitations:**

The limitations of NeuPRINT could receive a more thorough treatment (see points above).

---

> ### Author Rebuttal · Authors · 2023-08-10
>
> **“Is the boost in performance with NeuPRINT solely coming from the population activity and behavioral covariates?”**
>
> First, to address the reviewer’s concern on the role of population activity in the improved accuracy, we performed multiple new experiments to also add population activity to LOLCAT and its variants. (Please see the results in Table R1 in the overall response and Fig R1 in the attached pdf.) We reported the performance of the original implementation of LOLCAT (which became publicly available only after the initial submission) as ‘LOLCAT’. We also tuned LOLCAT with different architectures and formats of inputs. We changed its original attention module with a standard transformer architecture, and tested linear and multi-layer perceptron (MLP) classifiers, and different versions of inputs: spike interval distribution (ISI) (reported as ‘transformer+ISI’) v.s. raw fluorescence trace (reported as ‘transformer+raw’). To add population activity to these models, within the windows where ISI is computed, we calculated the mean and standard deviation of population activity in the center and surround partitions of each neuron, as well as the mean and standard deviation of behavior variables, and concatenated them with the ISI features. We also performed additional hyperparameter tuning for our NeuPRINT, LOLCAT and its variants to improve their performance. With these modifications, we evaluated both conditions (with and without population activity). The max subclass accuracy of the LOLCAT variants has improved from 0.474 to 0.561, and class accuracy from 0.608 to 0.680. However, these are still significantly worse than the performance of NeuPRINT, with 0.756 in subclass accuracy and 0.807 in class accuracy.  Interestingly, adding population activity to LOLCAT or its variants does not necessarily improve its accuracy. On the subclass prediction task, the best performance (0.561) is achieved from transformer + ISI without using population activity. For the class prediction task, adding population activity is slightly improving the performance from 0.664 to 0.680. While for NeuPRINT, adding population activity significantly improves the accuracy from 0.537 to 0.756 on subclass prediction.
>
> On the other hand, we emphasize that using population activity and behavior information to predict neurons’ transcriptomic identity instead of only using individual neuron’s activity alone is one of the major contributions of our work, regardless of whether this novel concept could be easily adapted to existing methods and boost their performance or not.
>
> **“Second, I am a little confused about the PCA/UMAP baselines. Here the authors selected a random window of time to project the data down to 64 components. Why 64? Did this account for a specific amount of variance? ...”**
>
> The PCA/UMAP baselines are provided merely to orient the reader. We do not expect those to ever outperform LOLCAT. Since the reported experimental sessions last for 5000 time steps, and we evaluate across 6 sessions, simply applying these baselines to the full time-series would create scalability issues. We chose the same setups and parameters for these baselines as those of NeuPRINT, so they would serve as naive unsupervised representation learning baselines, with a similar reconstruction framework, yet without a dynamical model and time-invariance constraint.
>
> **“Whether or not NeuPRINT outperforms LOLCAT, the LOLCAT paper systematically interrogates their model for insight into the underlying biology of neural systems.”**
>
> First, while a systematic interrogation is outside of scope (The LOLCAT paper was published in a life sciences journal. Our submission is oriented towards attendees of NeurIPS, primarily a machine learning venue.), our manuscript does reveal a novel neurobiological insight: Our results demonstrate that there is conditional information on the type of the cell of interest in the in-vivo population activities of cells other than the cell of interest. (e.g., Fig. 3: information in the population activity (or pupil size, etc) improves the accuracy of identifying the type of the cell, given the activity of that cell.)
>
> In addition, shown in Figure R2 and Table R2 in the attached pdf, we tested our model and LOLCAT and its variants on new experiments. We include recordings with 3 sets of stimuli (drifting gratings and 2 different natural scenes) in addition to the spontaneous activity recordings. We find that the predictive performance of NeuPRINT improves when more complex stimuli are used (i.e. subclass accuracy improved from 0.756 to 0.805, and class accuracy improved from 0.807 to 0.836 when switching the condition from spontaneous recordings to natural scenes), which indicates intricate, identity-specific dynamics can become more prominent when the underlying computation becomes more complicated. This is in line with and extends the findings of Bugeon et al. We thank the reviewer for this comment, and we will edit the manuscript to highlight this finding.
>
> **“The discussion of time-variant and time-invariant is fairly confusing. Are the authors saying that ISI distributions are time-variant?”**
>
> For LOLCAT, the ISI distribution is based on a 3-second trial sample. Therefore, the ISI distributions of neurons vary across trials, creating a time-varying descriptor. Compared to that, NeuPRINT only learns a fixed set of parameters (time-invariant) across all trials and all sessions. Importantly, this calculation scales well to long recordings because NeuPRINT has a dynamics model. We hope this explanation clarifies the confusion and we are happy to further elaborate and correct if needed.

---

> > ### Comment · Reviewer_MNFF · 2023-08-14
> >
> > I appreciate the comprehensive response from the authors.  I have updated my score accordingly.
> >
> > A follow-up question: this may just be my confusion, but in the original manuscript it appears that NEUPRINT w/o population statistics on subclass prediction (Table 3) performs worse than LOLCAT (Table 1). However, in the new attachment it looks like NEUPRINT performs better in all configurations, both in class and subclass prediction. Can the authors explain this discrepancy? Is it due to differences in how LOLCAT was implemented?

---

> > > ### Author Response · Authors · 2023-08-15
> > >
> > > We thank the reviewer for considering our response, improving their score, and the follow up question. Yes, the change is due to the differences in implementation. The current results for "LOLCAT" in the rebuttal are from their public implementation. In the original paper, the results “LOLCAT w/ ISI” and “LOLCAT w/ raw” are from our re-implementation of LOLCAT before their public implementation became available. We used the self-attention mechanism in transformer instead of LOLCAT original attention. Noticing the differences, we decided to rename our implementation to avoid confusion (“LOLCAT w/ ISI" becomes “Transformer+ISI”, “LOLCAT w/ raw” becomes “Transformer+Raw”). Since we are now able to add population inputs to all models, in Table R1, we compare all models across different architectures and input choices. We will add this clarification in the revision.

---

### Official Review · Reviewer_5dAW · 2023-07-04

**Soundness:** 3 good
**Presentation:** 3 good
**Contribution:** 2 fair
**Rating:** 5
**Confidence:** 3

**Summary:**

Understanding neural dynamics is crucial for studying brain function, but most existing methods assume that all neurons are functionally identical, which is not the case at the biological level. The biological variability among neurons can influence their dynamics, and since this variability is genetically encoded, the dynamics associated with neuron types would be time-invariant. This work introduces a method that uses transformers to learn a time-invariant representation of neural dynamics. The learned representation can reconstruct neural activity more accurately than existing methods when applied to neural recordings in mice. The authors also demonstrate that the genetic identity of the cell can be captured more robustly from this representation compared to alternative methods and lesion versions of the model.

**Strengths:**

The paper is adequately written and technically good. The method was tested and shown to work well when applied to calcium neural recordings in mice. The introduced method utilizes transformers to identify time-invariant dynamics associated with neural responses and relates them to the genetic heterogeneity observed in neural populations. Understanding the interplay between genetics and other factors, such as learning or external stimuli, in their relationship to neural dynamics is critical for studying neural function, and this work successfully extracts specific dynamics that can help predict neural type.

**Weaknesses:**

While the authors demonstrate how their method can extract neural dynamics related to neuron type, it is important to note that the dataset used only contains spontaneous neural activity. Neural dynamics are typically associated with specific computations and not just baseline states. Therefore, extracting time-invariant or baseline dynamics is relatively straightforward, and the reconstruction analysis may have limited significance. A more interesting application of these methods would be to disentangle dynamics associated with time-invariant or genetic factors from those associated with task variables when animals are engaged in a task or exposed to stimuli.
In terms of identifying neural type in the supervised learning task, the authors compare the classification performance of their method to other methods using neural dynamics. However, they overlook alternative approaches based on multiple neural properties such as firing rates, autocorrelograms, or morphology. If the goal of the introduced method is to label subpopulations or neurons, a direct comparison with commonly used approaches to putatively label neurons would be necessary to fully assess the significance of the method. Moreover, in the context of calcium imaging, the expression of the calcium indicator is typically given by genetic markers, so the type or types of neurons being monitored are known. Incorporating this explicit knowledge of neuron type when extracting dynamics could improve the accuracy of the method.
The accuracy results demonstrate that the proposed method can identify neuron types better than alternative methods. However, the overall accuracy is relatively low and fails to identify less prevalent neuron types. This suggests that the practical utility of the method for neuroscience research may be limited. Providing information on the computational and data demands of the method, as well as its applicability to other datasets, would be important to assess its relevance.

**Questions:**

Can the approach be directly used on other kinds of neural recordings like electrophysiological data?

**Limitations:**

The authors should include a section or clear description of the limitations and assumptions of the method. This section would also benefit from including information on the computational cost, training time, and data demands associated with the method.

---

> ### Author Rebuttal · Authors · 2023-08-10
>
> We thank the reviewer for their constructive review.
>
> **“the dataset used only contains spontaneous neural activity”**
>
> We believe this is not a limitation of the proposed method. Our method could be directly applied to neural recordings under different conditions, stimuli, and tasks. To address the reviewer’s concern on limited evaluation on spontaneous activity recordings only, we have performed multiple new experiments. We test our model based on parts of the Bugeon et al dataset that we have not used in the initial submission, thus representing completely new evaluations, including testing on recordings with 3 sets of visual stimuli (drifting gratings and two different natural scenes). Please see the tables in our overall response, Table R2, and Figure R2 in the attached pdf file. All of these new experiments still support our original findings that our model outperforms other baselines. Interestingly, we also found the predictive performance of NeuPRINT improved when more complex stimuli were applied (i.e. subclass accuracy improved from 0.756 to 0.805, and class accuracy improved from 0.807 to 0.836 when switching from spontaneous activity to natural scenes), which indicates that intricate, identity-specific dynamics can become more prominent when the underlying computation becomes more complicated. We will add the new results and discussions to the revised manuscript.
>
> **“overlook alternative approaches based on multiple neural properties such as firing rates, autocorrelograms, or morphology”**
>
> Alternative approaches based on activity (e.g., firing rates) are studied in detail in the LOLCAT paper and LOLCAT was found to be superior to those approaches. Since NeuPRINT outperforms LOLCAT. Therefore, it should outperform those alternative approaches. We thank the reviewer for this suggestion. We will revise the manuscript to mention such alternative approaches and cite the LOLCAT paper to suggest that those alternative approaches are not as powerful in in-vivo settings.
>
> On the other hand, cell type identification based on morphology is a very different problem. The reason is that there are no methods that record both in-vivo activity and neuronal morphology. Nascent studies combining calcium imaging with electron microscopy are upcoming, but these are limited by the much smaller field of view of electron microscopy. Therefore, while being the classical approach to identifying neuronal types, morphological analysis is not (yet) relevant for the purposes of studying in-vivo population activity in the context of neuronal identities. To address the reviewer’s concern, we will edit the Introduction to provide a morphological and historical perspective on neuronal cell types.
>
> **“the expression of the calcium indicator is typically given by genetic markers, so the type or types of neurons being monitored are known”**
>
> We believe this represents a key confusion and we hope that our overall response addresses this. Briefly, as the reviewer correctly mentions, the typical setup includes markers for, say, Pvalb neurons. Therefore, *only* Pvalb neurons are visible under the microscope. The experimenter already knows the cell type information, so there is nothing to infer on the recorded cells. But, what is the relationship between the activities of these Pvalb neurons and the other neurons? Those other neurons were never visible in such a typical setup and therefore cannot be studied.
>
> If, instead, a pan-neuronal marker is used, then potentially neurons of all types are visible under the microscope. Now, however, the experimenter does not know anything about cellular identity. This is the problem we address with NeuPRINT, with the help of emerging datasets that use pan-neuronal markers. Once inference accuracy becomes acceptable, post-hoc spatial transcriptomics will no longer be necessary (e.g., this will be recovered computationally by NeuPRINT).
>
> If the researcher is interested in, for instance, only the interactions between excitatory neurons in the mouse cortex, then the same argument can be repeated for pan-Glutamatergic markers. Conceptually, this will be identical to the problem posed here and NeuPRINT should provide a compelling solution as is.
>
> **“However, the overall accuracy is relatively low and fails to identify less prevalent neuron types”**
>
> First, our updated results show higher accuracy with more hyperparameter tuning, and under more complex stimuli (natural scenes) i.e. from 0.732 to 0.805 for subclass prediction, from 0.752 to 0.836 for class prediction.
>
> We agree that the existing accuracy values might not be high enough for NeuPRINT assignments to be trusted for individual cells. However, modern population recording experiments typically profile many cells. In these settings, NeuPRINT enables statistical inference and hypothesis testing. (e.g., is an event more likely to happen for excitatory or inhibitory neurons?)
>
> **“Information on the computational and data demands”**
>
> To address the reviewer’s concern, we will revise the manuscript to discuss computational and data demands of NeuPRINT. NeuPRINT has 833K parameters and takes 2h in total for training and inference on a single Nvidia Tesla V100 GPU. We have included details on the size of the dataset we used in Section 3.1 of the main paper.
>
> **“Can the approach be directly used on other kinds of neural recordings like electrophysiological data?”**
>
> If the reviewer means inference in, for instance, high-density electrode recordings using a pre-trained NeuPRINT model (based on calcium data), then, we do not anticipate this to work because the model takes advantage of the specifics of the recording modality as well as intrinsic relations, as discussed under Sec. 2. When such recordings with post-hoc spatial transcriptomic profiling become available, we expect NeuPRINT to be applicable and useful following training with such a dataset. We will add this perspective to the Discussion of the revised manuscript.

---

> > ### Comment · Reviewer_5dAW · 2023-08-15
> >
> > I thank the authors for their comprehensive response. I have updated my score accordingly.
> >
> > While I believe that the proposed methods has some limitations, it also paves the way for new insights. This is specially true if it can be applied to recordings when the network is engaged in a task.

---

> > > ### Author Response · Authors · 2023-08-15
> > >
> > > We thank the reviewer for considering our response and improving their score. Beyond the tasks of engaging with natural scenes (as we now demonstrate in the rebuttal), are there tasks involving visual processing that the reviewer would like us to discuss in a revision?

---

### Official Review · Reviewer_QXoV · 2023-07-05

**Soundness:** 3 good
**Presentation:** 3 good
**Contribution:** 2 fair
**Rating:** 6
**Confidence:** 2

**Summary:**

The authors evaluate several approaches for identifying neural types and subtypes (e.g., GABAergic cells are commonly divided into several subclasses Lamp5, Pvalb, Vip, Sncg, Sst). They used a dataset from a recently (2022) published neuroscience paper and compared their approaches to a recent study that had the same objective. To identify neural types, they use several 2-photon calcium imaging recordings (that provide a proxy for neural activity) from mice V1 and behavioral information (running speed and pupil size). They used mRNA expression for 72 genes to label the neurons according to the class (this was then used to compute the accuracy). While performance for class accuracy was relatively similar for all methods, the authors concluded that an approach based on the "Attention is all you need" transformer led to the best results classification results for subclass prediction. They evaluated the contribution of different factors of the model (e.g., running speed and pupil size) and concluded that each contributed to the classification accuracy.

**Strengths:**

The authors compared several approaches and provided detailed information about parameters in the supplemental material. They conducted an ablation study by evaluating the contribution of different pieces of data. The results are presented in a clear form, illustrating the overall accuracy as well as confusion matrices. The proposed approach led to better accuracy than in recent (2023) previous work that used the same dataset.

**Weaknesses:**

The evaluation was limited to a single dataset.

While, as the authors state in the discussion, this is an initial attempt to tackle this problem, it remains unclear whether the results would substantially improve if the authors did a hyperparameter sweep or used a more powerful transformer architecture.






**Questions:**

While the authors compared several approaches and concluded that the one based on transformers led to the best results, I wonder if they conducted hyperparameter sweeps to evaluate that the difference in the results between different methods was not because of different choices of a learning rate.

While table S3 was very useful for examining the hyperparameters that were used, it would be helpful to provide explicitly the number of trainable parameters used in each approach.

The authors used transformer architecture from "Attention is all you need" paper. Would more recent transformed architectures lead to better performance?

Was there a reason to use only 2 attention heads?

What was the robustness of the approach? For example, if you run the code twice with different seeds, how many cells would change the class/subclass?

Did different methods provide somewhat complementary results (e.g. were some neurons correctly labeled with recurrent model and incorrectly with transformer)? If yes, would it be possible to use some form of ensemble learning to get an even better performance?



**Limitations:**

The paper addresses limitations (although that could be expanded) and has a broader impact section.

---

> ### Author Rebuttal · Authors · 2023-08-10
>
> We thank the reviewer for their constructive review.
>
> **"The evaluation was limited to a single dataset."**
>
> We believe our general response (“Uniqueness of the dataset and study” and “Robustness of the method”) addresses the key points identified as weaknesses by this reviewer. Please refer to that response for clarifications and results of new experiments. Briefly:
> In that response, we argue that the Bugeon et al dataset is the only publicly available dataset that is suitable for the problem posed in this paper. To address the broader point, we will edit the manuscript to summarize the discussion above.
>
> To address the reviewer’s concern on limited evaluation, we have performed multiple new experiments. Some of these use parts of the Bugeon et al dataset that we have not used in the initial submission, thus representing new evaluations, including testing on a previously unseen animal, testing on recordings with 3 sets of stimuli (drifting gratings and 2 different natural scenes). Please see the tables in our overall response and the attached pdf file. All of these new experiments support our original findings, and will be added to the revised manuscript.
>
> **“It remains unclear whether the results would substantially improve if the authors did a hyperparameter sweep or used a more powerful transformer architecture.”**
>
> We have followed the reviewer’s suggestions to tune a set of hyperparameters that are critical for improving the models’ performance, please refer to new results Fig R1 and Table R1 in the attached pdf. Using a new set of hyperparameters (i.e. number of attention heads: 8,  embedding dim: 128, learning rate: $10^{-3}$) for our model NeuPRINT improved the accuracy substantially (subclass accuracy from 0.732 to 0.756, class accuracy from 0.752 to 0.807). At the same time, we also tried to improve our baseline LOLCAT by replacing its original simple attention structure with a more powerful self-attention mechanism from transformer, changing the linear classifier to multi-layer perceptron (MLP) classifier, and testing different versions of inputs: spike interval distribution (ISI) (refer to ‘Transformer+ISI’) v.s. raw fluorescence trace (refer to ‘Transformer+Raw’). We also performed additional hyperparameter tuning for these baselines. Subsequently, the subclass accuracy improved from 0.474 to 0.561, and class accuracy from 0.608 to 0.680, which are still worse than the performance of NeuPRINT.
>
> **“Number of trainable parameters used in each approach”**
>
> The numbers of trainable parameters used in each approach are as follows: LOLCAT: 8.5K, Transformer+ISI: 1.24M, Transformer+Raw: 1.19M, NeuPRINT: 834K.
>
> **“What was the robustness of the approach?”**
>
> We have conducted a sensitivity analysis in the supplement (Table S1), where the variance of using different random seeds seems relatively small. And we found our approach could also be generalized from a single animal to multiple animals as shown in Supp. Table S2. As mentioned in our general response, we also reported our methods with multiple new comparisons (see attached pdf), and established the robustness of NeuPRINT under different (and practically relevant) experimental setups, including testing on recordings with 3 sets of stimuli (drifting gratings and 2 different natural scenes). Furthermore, we also did a new cross-animal test where NeuPRINT and the downstream classifiers were trained with data from 3 animals. Any data from one animal was held out to be used for testing. At test time, we kept the dynamical model and downstream classifier fixed and only inferred the time-invariant representation for the unseen animal, and directly tested the accuracy for downstream tasks. Interestingly, NeuPRINT generalizes well in this setting, achieving subclass prediction accuracy of 0.586 and class prediction accuracy of 0.804. These new experiments support our original findings. We will incorporate them into the revision.
>
> **“If you run the code twice with different seeds, how many cells would change the class/subclass?”**
>
> To address the reviewer’s concern, we add the following set of results on the average proportion of cells changing their assignments across runs to the revision: for two NeuPRINT transformers using the same hyperparameters but different random seeds (with class accuracies 0.814 and 0.826), 16.7% of their predictions are not matched: model 1 is incorrect, model 2 is correct (8.9%), model 1 is incorrect, model 2 is correct (7.7%). For subclass prediction, their accuracies are 0.732 and 0.746, and there is 27.8% mismatch in their prediction: model 1 is incorrect, model 2 is correct (14.1%), vice versa (12.7%).
>
> **“Did different methods provide somewhat complementary results?”**
>
> The reviewer raises an interesting question on the potential to use ensemble learning. We agree that this is potentially a fruitful approach. As a starting point, we performed a mismatch analysis for the results from the recurrent network and the transformer. Particularly for subclass prediction, when the transformer has an accuracy of 0.731, and the RNN has an accuracy of 0.571, there is a large mismatch (54.6%), suggesting there might be benefits to performing ensemble learning. We will add an ensemble study to the revision, where we will train a classifier based on concatenating NeuPRINT representations from both transformer and recurrent network.

---

> > ### Comment · Reviewer_QXoV · 2023-08-15
> >
> > I thank the Authors for detailed response. I think additional experiments add value to the work. I updated my score accordingly.

---

> > > ### Author Response · Authors · 2023-08-15
> > >
> > > We thank the reviewer for considering our response and improving their score.

---

### Author Rebuttal · Authors · 2023-08-10

We thank the reviewers for their detailed comments. We would like clarify two key points that we thought are at the heart of much confusion:

**Uniqueness of the dataset and study**: Population recordings with calcium imaging are typically performed on mouse lines that are selective for a subset of neurons (e.g., Pvalb neurons). The large datasets produced by the Allen Institute are prominent examples of this approach. While the advantage is that the researcher knows that the recorded neurons are Pvalb neurons, the obvious disadvantage is that only Pvalb neurons are visible in this setting. Hence, interactions between different neuron types, concurrent responses of different neurons to stimuli, etc cannot be studied. Similarly, the information that other neurons carry on the type of a neuron of interest cannot be properly studied with these recordings because only one kind of neuron is available. Finally, classifying the neurons in these datasets is not a problem of practical interest. (e.g., There’s no utility to predicting that the neuron is of type Pvalb when we already know that all the neurons identified in that experiment are of type Pvalb.)

A recent method which pairs calcium imaging with post hoc spatial transcriptomics opens new avenues by allowing experiments with pan-neuronal virus infections (recording from all kinds of neurons). This is because the identity of the neurons recorded by the calcium imaging session will be subsequently identified by the spatial transcriptomics experiment. As of today, the dataset by Bugeon et al is the only such publicly available dataset. This underlines a challenge: spatial transcriptomics is slow, expensive, and not available to many research labs. This is the problem that our study aims to address: by using the available calcium imaging+spatial transcriptomics data as a dictionary, NeuPRINT aims to predict the cell types in calcium imaging-only experiments.

**Robustness of the method**: We would first like to apologize for not referring to the supplement more prominently in the initial submission. Table S1 includes a sensitivity study over five random seeds, which supports all of the original findings, and Table S2 extends our models to test on multiple animals.

Our original use of spontaneous recordings in our demonstration was not motivated by accuracy considerations. We chose those recordings in the reported experiments because (i) spontaneous recordings are performed almost ubiquitously in the field, enabling immediate practical utility, (ii) such recordings are prominently available in the Bugeon et al dataset. Indeed, it is perhaps likely that more complicated stimuli or tasks will require more complicated neuronal computations and reveal intricate, type-specific computations, thus increasing the accuracy of cell type identification.

In order to illustrate on this point, and also demonstrate the robustness of our approach. We have performed multiple new experiments, and tested our model based on parts of the Bugeon et al dataset that we have not used in the initial submission, including testing on recordings with 3 sets of visual stimuli (drifting gratings and two different natural scenes), following the same reporting protocol. The summary table below suggests an increase in cell type identification accuracy from in-vivo activity as stimulus complexity increases, which is a novel insight:

|      NeuPRINT     | Spontaneous | Drifting Grating | Natural Scenes 1 | Natural Scenes 2 |
|:-----------------:|:-----------:|:----------------:|:----------------:|:----------------:|
| Subclass Accuracy |    0.756    |       0.780      |       0.805      |       0.782      |
|   Class Accuracy  |    0.807    |       0.809      |       0.836      |       0.836      |

We will incorporate a more detailed table including these results (see attached pdf) to the revision.

---

### Decision · Program_Chairs · 2023-09-21

**Decision:**

Accept (poster)

**Comment:**

This paper proposed an original method for predicting neurons' transcriptomic identity from physiological recordings, which generated a high level of interest and discuss amongst the reviewers. I congratulate the authors for their detailed replies to reviewer comments, which were critical in convincing them to improve their assessment of the paper's contributions and raise their scores. Although none of the reviewers moved far enough to strongly endorse the paper for acceptance, they were unanimous in rating it as "above threshold".  In light of this agreement, I am inclined to agree that it should be accepted. Congratulations! Please attend carefully to reviewer comments and suggestions when revising the final manuscript.